# Clinical and Functional Characteristics of the E92K *CFTR* Gene Variant in the Russian and Turkish Population of People with Cystic Fibrosis

**DOI:** 10.3390/ijms24076351

**Published:** 2023-03-28

**Authors:** Elena Kondratyeva, Yuliya Melyanovskaya, Nataliya Bulatenko, Ksenia Davydenko, Alexandra Filatova, Anna Efremova, Mikhail Skoblov, Tatiana Bukharova, Viktoriya Sherman, Anna Voronkova, Elena Zhekaite, Stanislav Krasovskiy, Elena Amelina, Nika Petrova, Alexander Polyakov, Tagui Adyan, Marina Starinova, Maria Krasnova, Andrey Vasilyev, Oleg Makhnach, Rena Zinchenko, Sergey Kutsev, Yasemin Gokdemir, Bülent Karadag, Dmitry Goldshtein

**Affiliations:** 1Research Centre for Medical Genetics, 1 Moskvorechye St., 115552 Moscow, Russiaanna.efremova.83@gmail.com (A.E.);; 2Pulmonology Scientific Research Institute, Federal Medical and Biological Agency of Russian Federation, Orekchovy Boulevard, 28, 115682 Moscow, Russia; 3Division of Pediatric Pulmonology, Marmara University, 34722 Istanbul, Turkey

**Keywords:** cystic fibrosis (CF), CFTR, E92K, CF Registry, intestinal current measurements (ICM), intestinal organoids, forskolin-induced swelling (FIS) assay, VX-809 corrector, VX-770 potentiator, genotype–phenotype correlation

## Abstract

The pathogenic variant E92K (c.274G > A) of the *CFTR* gene is rare in America and Europe, but it is common for people with cystic fibrosis from Russia and Turkey. We studied the effect of the E92K genetic variant on the CFTR function. The function of the CFTR channel was studied using the intestinal current measurements (ICM) method. The effects of CFTR modulators on the restoration of the CFTR function were studied in the model of intestinal organoids. To assess the effect of E92K on pre-mRNA splicing, the RT-PCR products obtained from patients’ intestinal organoid cultures were analyzed. Patients with the genetic variant E92K are characterized by an older age of diagnosis compared to homozygotes F508del and a high frequency of pancreatic sufficiency. The results of the sweat test and the ICM method showed partial preservation of the function of the CFTR channel. Functional analysis of *CFTR* gene expression revealed a weak effect of the E92K variant on mRNA-CFTR splicing. Lumacaftor (VX-809) has been shown to restore CFTR function in an intestinal organoid model, which allows us to consider the E92K variant as a promising target for therapy with CFTR correctors.

## 1. Introduction

Cystic fibrosis (CF) is an autosomal recessive disorder caused by variants in the cystic fibrosis transmembrane conductance regulator gene (*CFTR*). The incidence of the disease in Russia is 9–10 per 100,000 newborns, and in Turkey, it is 2.9 per 10,000 live births [1,2]. The *CFTR* gene is located on chromosome 7, on the long arm at position q31.2 (7q31.2), has a length of about 189,000 base pairs, and includes 27 exons. The gene product is an ion channel that transports chloride ions through the membranes of epithelial cells and consists of 1480 amino acid residues [2]. The CFTR protein is a member of the superfamily of ATP-binding cassette (ABC) transporters. CFTR consists of five domains: two membrane-spanning domains (MSD1 and MSD2), which form a channel of chloride ions, two nucleotide-binding domains (NBD1 and NBD2), which bind and hydrolyze ATP, and a regulatory (R) domain-containing site’s phosphorylation. Dysfunction of the CFTR channel in the epithelial cells of target organs blocks the transport of chlorine ions and increases the absorption of sodium ions, resulting in a decrease or complete cessation of fluid secretion through the apical membrane of epithelial cells, which leads to various pathogenetic events in various organs [3]. Clinically, cystic fibrosis (CF) is characterized by chronic and progressive damage of the respiratory tract, chronic exocrine insufficiency of the pancreas, pathology of the gastrointestinal tract and liver, as well as a decrease in reproductive function in patients, and a number of other manifestations [4]. Complex CF therapy is aimed at eliminating the causes and symptoms of the disease, which leads to an increase in the life expectancy of patients (currently, the average age of patients in Russia is 13.7 ± 9.7 years [5]). Currently, a pathogenetic approach to the treatment of the disease has been developed—a direct effect on the CFTR protein through CFTR modulators and an active search is underway for new molecules capable of stimulating the synthesis, transport, or functional activity of the defective CFTR protein [6]. The possibility of a therapeutic effect on the protein is determined by the class of pathogenic variant CFTR. Depending on the mechanism affecting the function of the protein, all genetic variants are divided into seven classes. Variants of classes I-III and VII cause a more severe course of the disease, whereas with classes IV-VI, partial operation of the ion channel may persist.

To date, more than 2000 variants of the *CFTR* gene have been noted in the CFTR1 database [7]. According to the data of the European Cystic Fibrosis Society Patient Registry (ECFSPR) for 2020, seven pathogenic variants of the *CFTR* gene are frequent in the European cohort of CF patients, and the relative frequencies of each of which are ≥1% F508del (60.41%), G542X (2.75%), N1303K (2.18), G551D (1.26%), W1282X (1.07%), 2789 + 5G→A (1.07%), and 3849 + 10kbC→T (1.0%) [8]. The relative proportions of each of the 11 more variants account for ≥ 0.5% of the total number of mutant CFTR alleles tested. At the same time, in some populations, these variants can reach significant frequencies, so the frequency of variant 2183AA→G in the population of Armenia is 10.0%, and the variant R1162X in Slovenia is 5.0% [8]. The remaining variants with an allele frequency of < 0.5% total 22.5% of all examined CFTR alleles. Most of them are rare or even unique, i.e., found in a small group of patients or even in a single family [8]. In a variety of populations, any one specific variant may predominate in the spectrum, for example, c.3276C > G (Y1092X) in Cameroon, c.3310G > T (E1104X) in Tunisia, and и c.3700A > G (I1234V) in Qatar and in certain Arab tribes [9]. Therefore, the distribution and frequency of variants of the *CFTR* gene vary significantly in different countries and ethnic groups. According to the Russian Registry of Patients with Cystic Fibrosis in 2020, out of 230 identified pathogenic variants, 11 with relative proportions exceeding 1% are frequent in Russian patients: F508del (52.61%), CFTRdele2.3 (6.15%), E92K (3.25%), 1677delTA (2.12%), 3849 + 10kbC > T (2.11%), 2143delT (2.02%), 2184insA (1.93%), W1282X (1.73%), L138ins (1.53%), N1303K (1.53%), and G542X (1.46%) [5]. In the Russian Federation, the E92K variant (c.274G > A) is more common in the Turkic-speaking populations of the Volga-Ural region with a maximum frequency in the Chuvash population (55%). According to the data of the National Registry of Patients with Cystic Fibrosis of Turkey for 2020, the frequency of variants is listed as F508del (22.5%), N1303K (4.7%), 2789AA- > G (4.1%), G542x (4.1%), 1677delTA (4.1%), E92K (2.7%), and G85E (2.6%) [10]. In the GnomAD database [11], E92K is not registered. In the ClinVar database, variant E92K (c.274G > A) is defined as pathogenic [12]. In the CFTR2 database, the E92K variant is defined as leading to cystic fibrosis, pancreatic insufficiency was noted in 53% of patients carrying this variant in a compound with other *CFTR* gene variants leading to impaired pancreatic function, its frequency is 0.00034 [13].

The intestinal organoids model allows personalized evaluation of the effectiveness of CFTR modulators [14]. Intestinal organoids are closed self-organizing 3D structures with an apical membrane facing inside the cavity (lumen). Treatment of intestinal organoids with forskolin causes cAMP-dependent activation of the CFTR channel and their swelling. The response of organoids to stimulation with forskolin directly depends on the function of CFTR [15,16,17]. This principle is the basis of a test called CFTR-dependent forskolin-induced swelling (FIS), which allows the personalized study of the residual activity of the CFTR chloride channel in CF patients and predicts an individual response to therapy with CFTR modulators aimed at restoring the function of mutant CFTR.

The aim of this study is to investigate the dysfunction of the CFTR channel in patients with CF from the Russian Federation and Turkey, who have these countries’ pathogenic variant E92K genotype, are subjected to the frequency of this variant, and experience the genotype–phenotype correlation.

## 2. Results

### 2.1. Assessment of the Frequency of the Genetic Variant of E92K in Different Regions of the Russian Federation and in Turkey

An analysis of the Russian Registry of patients with cystic fibrosis for 2020 showed that E92K ranks 3rd in the frequency of occurrence (3.25% of the allelic frequency of all genetic variants) [5].

The allelic frequency of the E92K variant in the federal districts of the Russian Federation is shown in Figure 1, where it can be seen that the highest proportion of the prevalence of this variant in the Volga Federal District is 8.6%, in second place is the North Caucasus Federal District—7.5% [5]. Moreover, in Chuvashia (which is part of the Volga Federal District) in 2020, the allelic frequency of E92K was 50%.

In the Turkish Cystic Fibrosis Patient group, the E92K variant frequency in 2020 was 2.7%.

### 2.2. Characteristics of the Clinical Picture of Patients with the Genetic Variant E92K

The clinical characteristics of patient groups, including the E92K genetic variant, are presented in Table 1. The age at the time of the study for the E92K/I class group was similar to that of the F508del/F508del homozygote group, and the patients with the E92K/E92K, E92K/F508del genotype were older, and the E92K/IV-VI classes patients had the highest indicators of age at the time of examination. Similar results were obtained regarding the age of diagnosis. The study of sweat test indices revealed that the test indices were higher in the F508del/F508del group in comparison with other groups. The lowest rate was registered in the E92K/IV-VI classes group (Table 1).

Table 1 presents data on pancreatic insufficiency in the studied genotypes. Fecal elastase-1 was decreased in 90.2% of F508del/F508del patients compared to 22.7% of E92K/F508del patients, 25% of E92K/E92K patients, 40% of E92K/I class patients, 20% of E92K/IV-VI classes patients (*p* < 0.001).

Nutritional status (BMI) significantly differed in groups E92K/F508del (Me—17.2) and F508del/F508del (Me—15.8) (*p* = 0.011) (Table 1). There was no difference in lung function (FEV1 and FVC). Analysis of the studied bacterial pathogens of the respiratory tract showed that significant differences were observed in patients with MRSA, in the E92K/IV-VI classes group had the largest percentage of patients with this pathogen (25%) (*p* = 0.008), while in the E92K/E92K group—9.4%, E92K/F508del—7.3%, F508del/F508del—3.5%, and in the E92K/I class group, this pathogen was not detected at all.

A comparison of the complications of the disease revealed that hemoptysis is observed in 4.9% of patients in the E92K/F508del group in comparison with patients with the F508del/F508del genotype, in whose group this complication occurred only in 0.9% (*p* = 0.006). There were no significant differences in other complications. Amyloidosis was not registered in the study groups and data are not shown in Table 1.

The analysis of the therapy performed in patients with different genotypes revealed that the patients of the E92K/E92K group received less often inhalation of a hypertonic NaCl solution (*p* = 0.010), inhaled (*p* = 0.028) and oral antibiotics (*p* = 0.010) than in the F508del/F508del group. Patients with the E92K variant in the genotype received pancreatic enzymes significantly less frequently than patients in Group 5 (*p* < 0.001). A similar situation was observed with regard to the intake of ursodeoxycholic acid. Patients in the E92K/E92K, E92K/F508del, and E92K/IV-VI classes received it less often (75%, 80.5%, and 62.5%, respectively) than in the F508del/F508del group—93.5% (*p* < 0.001). Fat-soluble vitamins were prescribed less frequently to patients with genotypes E92K/E92K (65.6%) and E92K/F508del (76.3%). Low consumption of pancreatic enzymes, ursodeoxycholic acid, and fat-soluble vitamins is associated with the intact function of the pancreas in carriers of the E92K genetic variant (Table 1).

Thus, the analysis of the data from the Registry of the Russian Federation and Turkey for 2020 showed that the genetic variant of E92K belongs to the “mild” one, which is manifested by the preservation of pancreatic function and low conductivity values during the sweat test. In terms of lung function indices, no differences from the group of homozygotes F508del were obtained, but the development of such complications as hemoptysis was noted more often in the E92K/F508del group.

### 2.3. Evaluation of the Functional Activity of Ion Channels on the Surface of the Intestinal Epithelium by the Intestinal Current Measurements (ICM) Method

The analysis of the CFTR chloride channel function was investigated using the ICM method. The quantitative composition and genotypes of the study participants are presented in Appendix A.

Patients according to their genotype were divided into groups: Group 1—E92K/E92K, Group 2—E92K/I class, Group 3—E92K/F508del, and Group 4—E92K/IV-V class. Comparison Group 5, according to European recommendations [18] included three patients with the F508del/F508del genotype, and healthy with the CFTR (*wt*/*wt*) functional channel in Group 6.

The results of the ICM were shown in Table 2 and Figure 2. The short-circuit current density (ΔISC) in response to the introduction of amiloride (stimulation of sodium channels) was in Group 1—7.81 ± 3.83 µA/cm^2^, in Group 2—5.92 ± 1.01 µA/cm^2^, in Group 3—20.21 ± 7.29 µA/cm^2^, in Group 4—4.83 ± 0.39 µA/cm^2^, in Group 5—18.39 ± 5.62 µA/cm^2^, and in the group with functional channel CFTR—8.98 ± 2.23 µA/cm^2^.

The change in ΔISC in response to the introduction of forskolin/IBMX (stimulation of chlorine channels) was as follows: in Group 1—(E92K/E92K) 7.13 ± 1.05 µA/cm^2^, in Group 2—(E92K/I class)—5.67 ± 1.25 µA/cm^2^, in Group 3—(E92K/F508del) 17.5 ± 0.97 µA/cm^2^, in Group 4 (E92K/IV-V classes)—10.17 ± 2.34 µA/cm^2^, in Group 5 (F508del/F508del)—3.06 ± 0.89 µA/cm^2^, and in the control group (healthy control)—25.78 ± 3.37 µA/cm^2^. These results for patients with E92K differed significantly from the indicators of the homozygote group according to the genetic variant F508del and indicated the preservation of the residual function of the chloride channel.

In response to histamine administration, the ΔISC changes in the negative direction, which reflects the outflow of potassium ions from cells in cystic fibrosis. At the same time, the current density was in Group 1—11.63 ± 2.26 µA/cm^2^, in Group 2—33.5 ± 11.11 µA/cm^2^, in Group 3—10.14 ± 3.02 µA/cm^2^, in Group 4—26.08 ± 5.77 µA/cm^2^, in Group 5—21.5 ± 5.46 µA/cm^2^, and in the control group—101.68 ± 9.78 µA/cm^2^. (Table 2, Figure 2).

### 2.4. Evaluation of the Effect of Targeted Drugs on the Functional Activity of the CFTR Channel on Models of Intestinal Organoids

Due to the CFTR function (which in organoids is located in the apical membrane similar to the intestinal mucosa), water and chloride ions are transported inside the lumen, so that the organoids of a healthy person (*wt*/*wt*) are rounded and thin-walled, with a characteristic lumen (Figure 3).

Organoids cultures were obtained from each of the 10 study participants whose genotype revealed a mutant E92K-CFTR variant, of which 3 patients with the E92K/E92K genotype, 2 patients with the E92K/I class genotype (E92K/2143delT, E92K/CFTRdel1-11), 3 patients with the E92K/F508del genotype, and 2 patients with genotype E92K/IV-V class (E92K/L138ins, E92K/3849 + 10kbC→T) (Appendix A). The morphology of intestinal organoid cultures in each of the four patient groups does not differ from the F508del/F508del control with a non-functional CFTR channel: the lumen is reduced, the walls are thickened, and the organoids are not spherical (Figure 3).

For each patient, the effect of the CFTR corrector VX-809 and the CFTR potentiator VX-770 was evaluated on the intestinal organoid models.

It has been shown that FIS of intestinal organoids correlates with the severity of the course of the disease, in patients from groups 1–3, the response to forskolin at concentrations of 5 or 0.128 μM separately and together with the potentiator VX-770 is absent or very weak (Figure 4 and Figure 5). The obtained data indicate the almost complete absence of a functional CFTR channel in CF patients who are E92K homozygotes or heterozygotes with pathogenic variants 2143delT, CFTRdel1-11, F508del on the second allele. For organoid cultures obtained from Group 4 patients (the second variant was classified as “mild” and assigned to class IV or V) by 60 min, there was a pronounced response to forskolin, as well as to forskolin together with a potentiator (especially for the E92K/L138ins genotype), which allowed us to conclude that the functional protein CFTR is present on the surface of the apical membrane of the cells (Figure 4 and Figure 5, Appendix A). The observed effects are due to the contribution of a “mild” variant in the genotype. When using the VX-809 corrector, significant restoration of the functional CFTR protein occurs in all 10 organoid cultures from all experimental groups, while the VX-770 potentiator, when stimulated by 5 μM with forskolin, on average by 30% (groups 1–3), additionally enhances the effect of VX-809 (Figure 5).

### 2.5. Functional Assessment of the Effect of E92K Variant on Splicing

To assess whether an E92K variant affects splicing, we performed RT-PCR analysis of mRNA obtained from intestinal organoids culture of healthy control and E92K homo- and heterozygous patients (E92K/E92K and E92K/F508del, respectively). To rule out the possible degradation of the misspliced transcript by nonsense-mediated mRNA decay, we performed NMD inhibition on a heterozygous E92K/F508del patient’s intestinal organoids culture followed by mRNA extraction and RT-PCR. The electrophoresis of all obtained PCR products did not reveal any changes in the length as compared to the wild type which was confirmed by Sanger sequencing (Figure 6A). In the case of a heterozygous E92K/F508del patient sequencing showed that both alleles carrying CFTR variants are expressed at the same level. Furthermore, Sanger sequencing revealed no difference between PCR products obtained from intestinal organoids culture before and after NMD inhibition (Figure 6A).

Since minor splicing events may not be detected by routine RT-PCR followed by Sanger sequencing, we performed targeted next-generation sequencing of the PCR products obtained from healthy controls, E92K homozygous patients, and healthy carriers of the E92K variant (E92K/WT) (Figure 6B). The analysis of the homozygous patient-derived sample revealed about 6.8% of the mutant isoform with skipping of the exon 3. In addition, approximately 3.5% of the isoform with exon 3 skipping was found in E92K carrier-derived sample, whereas in *wt*/*wt* samples, we observed only approximately 0.3% of exon 3 skipping (Figure 6C).

## 3. Discussion

The pathogenic variant E92K was first described in 1993 by Nunes V et al. [19], leads to the replacement of glutamic acid with lysine, and is a missense variant in the MSD1. Later, it was found that E92K in some cases can affect splicing in patients with this variant in a homozygous state [20].

The genetic variant E92K was previously described mainly in the Turkic peoples [21]. In the Russian registers of patients with CF, E92K ranks 3rd in the frequency of occurrence among other pathogenic alleles: from 2.65% in 2011 [22] to 3.25% in 2020 [5]. In 2012, it was shown for the first time that the proportion of this genetic variant among patients with CF in the Chuvash Republic is 55.5%, which is the highest frequency of the E92K variant in the world since in other countries, this indicator does not exceed 0.1% [23,24]. In 2020, the allele frequency of E92K in Chuvashia was 50%.

According to the data of the National Registry for patients with cystic fibrosis in Turkey in 2020, the allele frequency of the E92K variant was 2.7%, ranking 7th [10].

The high prevalence of the pathogenic variant E92K in Russia (3rd place) and Turkey (7th place), allowed us to combine data and study the geno-phenotypic features of patients with the genotype E92K. E92K is more often referred to as a Class II pathogenic variant of CFTR, in which the folding or transport of protein to the apical membrane of cells is disrupted, and, as a consequence, its proteolysis occurs in the endoplasmic reticulum [25]. The article [26] shows that the amount of mRNA in the genetic variant of E92K is increased, but the amount of mature functional protein formed does not exceed 2%. Replacement of glutamic acid with lysine leads to a defect in the salt bridge in MSD1, which is necessary for folding CFTR [27]. At the same time, some authors refer to E92K as a pathogenic variant of class IV-V [28]. Russian studies have shown that in patients homozygous and heterozygous for the E92K variant, the function of the pancreas is preserved, and therefore genotypes with the E92K variant were classified as “mild” [29]; meanwhile, there is information on the CFTR2 international project that 53% of patients with E92K have pancreatic insufficiencies [13]. According to the data of patients with CF in the Russian Federation and Turkey (2020), pancreatic insufficiency developed in 25% of E92K homozygotes, in 23.7% of patients with the E92K/F508del genotype, and in 20% of patients with the E92K/IV-VI class genotype, which is significantly lower than indicated in the CFTR2 database. The preservation of pancreatic function in most patients with the E92K variant suggests that this variant is classified as a “mild” class; however, an analysis of the Registers of the Russian Federation and Turkey showed that lung damage in this category of patients does not differ in severity from the group of homozygotes, according to F508del. The largest number of patients who were newborn screened belongs to the group of F508del homozygotes, due to the fact that they are younger compared to groups I–IV, and screening in the Russian Federation began in 2007.

There were no differences in the frequency of infection of the respiratory tract by the main microbial pathogens and in the number of complications. It should be noted that the age of the patients with the genetic variant E92K, especially in the group E92K/IV-VI classes, is higher compared to the group of homozygotes according to F508del. Perhaps it is the increase in the age of patients that eliminates possible differences in lung function and complications.

The results of a sweat test to assess the function of the chloride channel in patients with the E92K variant in the genotype found that the sweat conductivity indicators of patients with the studied variant in the homozygous state were lower than in the F508del/F508del group. The ICM method also showed that for all genotypes with the E92K variant, the function of the CFTR channel is partially preserved, which corresponds to the clinical picture and literature data [20].

The results obtained on intestinal organoids correlate with the data provided by Ramalho and co-authors in the article 2021 [30] for a patient with genotype E92K/1811-1.6KbA > G. In a study on 10 patients with different genotypes, we showed that the E92K variant causes serious violations of the CFTR channel function. If a “severe” variant of class I is present in the genotype, there is a decrease in responses to all CFTR modulators, and, conversely, “mild” variants in combination with E92K levels the responses. This result may indicate that each allele of the *CFTR* gene affects the total amount and functional properties of the CFTR protein in a cystic fibrosis patient. We have shown that organoids with the same genotypes (three cultures with E92K/E92K and three cultures with E92K/F508del) demonstrate the variability of responses. For example, one of the three E92K/F508del cultures was characterized by a maximum response to 5 μM of forskolin after the action of the corrector VX-809 (patient 3). The results for the control F508del/F508del cultures correlate with the literature data [16,31] and the results of our own research [32]: the corrector VX-809 and the potentiator VX-770, when combined, restore the CFTR function, while each of them acts weakly independently. In genotypes with the E92K variant, the swelling of organoids in response to the use of VX-809 and VX-809 + VX-770 exceeds the results obtained for control by 3–8 and 1.5–3 times more, respectively, which makes it possible to consider the E92K variant as a promising target for therapy with CFTR correctors. Previously, Ren H. with co-authors on HEK293 cells transfected with pcDNA3.1(+)-CFTR plasmid. It was shown that VX-809 effectively restores the function of the CFTR channel in the presence of pathogenic variants in MSD1, including E92K [27]: when exposed to VX-809, a dose-dependent correction of E92K-CFTR folding disorders was observed to a normal level, and F508del-CFTR was adjusted to ∼15% of the *wt*/*wt* level. The results of our study are consistent with these data. Responses to VX-770 on intestinal organoids with the E92K variant were comparable to controls, except for the E92K/I class group, in which there was no response. Thus, in this study, it was shown that the genetic variant of E92K causes serious violations of the function of the CFTR channel, while the corrector VX-809 effectively restores it.

The first functional studies of missense variant E92K showed that, as the F508del, it causes incorrect protein folding, and, as a result, the misfolded channel remains in the cytoplasm and is absent in the cell membrane [25]. However, since this substitution is located at the first nucleotide of the *CFTR* exon 4, it has been suggested that the E92K variant may also affect pre-mRNA splicing. To assess the effect of this variant on splicing and to find out the exact molecular mechanism of its pathogenicity, we performed a functional analysis of the E92K variant. Targeted next-generation sequencing of PCR-product obtained from intestinal organoids culture mRNA showed that the E92K variant had a mild effect on splicing, leading to exon 3 skipping. These results are in agreement with the data, which were previously obtained using minigene systems [33]. We also tested whether the E92K variant can affect the expression level of the *CFTR* gene. Using an overexpression system, it was previously reported that the E92K variant can increase the expression of the *CFTR* gene several times in human cell lines [26]. Deep sequencing of RT-PCR products obtained from intestinal organoids of healthy controls and E92K homozygous patients did not reveal any differences in the expression of alleles carrying the E92K variant and F508del variant in heterozygous patients. Thus, it can be assumed that despite the weak influence of the E92K variant on splicing the main mechanism of pathogenicity of the E92K variant is the effect of missense substitution on the correct folding of the CFTR protein, which leads to degradation of the protein and its absence on the cell membrane.

The data obtained on intestinal organoids are at odds with clinical data and the results of functional tests (sweat test and ICM), which showed that the CFTR channel is partially functioning. We assume that these differences may be due to several reasons. Firstly, the methods of evaluating the function of CFTR by means of three methods differ significantly: a sweat test is carried out directly with the participation of the patient, the ICM method is performed on ex vivo rectal biopsies, and the forskolin test is performed in vitro on intestinal organoids. The conductivity of sweat is an indirect indicator of salt absorption in sweat ducts. The parameters obtained by the ICM method can be considered more reliable since they are the result of a direct measurement of the transmembrane current of chlorides mediated by CFTR [34]. The method of intestinal organoids involves a long (at least 2 weeks) cultivation of organoids before the forskolin test, during which the level of gene expression and protein synthesis, including regulatory ones, may change. Secondly, the effect of forskolin may be different in the epithelium of the intestinal biopsy and the organoid model. Thirdly, it is impossible to exclude the possibility of tissue-dependent splicing of mRNA CFTR in pancreatic tissue and intestinal epithelium, which can lead to the formation of functional CFTR in an amount sufficient to preserve the function of the pancreas.

Some variants of the *CFTR* gene are difficult to attribute to any of the known six classes of variants since they lead to different defects in the synthesis of the CFTR protein. Currently, the classification of genetic variants of the *CFTR* gene is being revised. The new classification is reflected in the Venn diagram and includes 31 classes of variants, including the original classes I, II, III/IV, V, and VI, as well as 26 combinations of them. The pathogenic variant of E92K, widespread in Russia and Turkey, also causes difficulties in assigning it to one or another class. The authors of the newly expanded classification refer to E92K as both II and III classes [35]. In our study, the phenotypic manifestations of the E92K variant (preservation of pancreatic function) allow us to attribute it to “mild” variants, but the results of the forskolin test on organoids (a strongly reduced response to forskolin, a weak effect of the CFTR ivacaftor potentiator on the function and a strong influence of the lumacaftor corrector on the restoration of CFTR function) allow us to attribute it to Class II. According to the conducted studies, it can be assumed that the genetic variant of E92K may belong to classes II–V. Class V is characterized by pathogenic variants of the CFTR gene with a significant decrease in the amount of CFTR protein on the plasma membrane.

## 4. Materials and Methods

### 4.1. Criteria for Inclusion of Patients in the Study. Description of the Clinical Picture

The study was conducted after the representatives of the patients signed an informed voluntary consent to participate in the study with the use of ICM and FIS assay in intestinal organoids. The study and the form of informed voluntary consent were approved by the Ethics Committee of the “RCMG” of the Ministry of Education and Science of the Russian Federation on 15 October 2018 (the chairman of the Ethics Committee is Prof. L. F. Kurilo).

In order to assess the features of clinical manifestations of the genetic variant E92K, data from the Russian and Turkish registers of patients with cystic fibrosis for 2020 were analyzed. The parameters were evaluated according to the Russian and European registers of patients with CF [36]. The study of pheno-genotypic relationships was carried out when comparing five groups of patients with CF with different genotypes: E92K/E92K (Group 1—33 individuals), E92K/I class (Group 2—36 individuals), E92K/F508del (Group 3—82 individuals), E92K/IV-V classes (Group 4—8 individuals), and F508del/F508del (Group 5—881 individuals) (Table 3).

The following parameters were considered: patient age, age at diagnosis, sweat test parameters, body mass index (BMI; kg/m^2^), and spirometric parameters, namely, forced expiratory volume in 1 s (FEV_1_) and forced vital capacity (FVC) of the lungs, presence of microorganism colonization in the bronchopulmonary system (*Pseudomonas aeruginosa, Staphylococcus aureus, MRSA*, *Burkholderia cepacia complex*, *Achromobacter spp*, *Stenotrophomonas maltophilia*, non-tuberculosis mycobacteria, other gram-negative microflora), pancreatic insufficiency, complications (meconium ileus, diabetes, osteoporosis, allergic bronchopulmonary aspergillosis (ABPA), etc.), and treatment.

The state of lung function was analyzed according to the data of the forced vital capacity of the lungs (FVC, %) and the volume of forced exhalation in 1 s (FEV1, %). Studies were conducted in accordance with the ERS/ATS criteria in a group of children who were able to perform a respiratory maneuver during spirometry [37].

The pancreatic function was assessed based on the level of fecal elastase-1 (FE-1). Levels of FE-1 below 200 μg/g stool suggested a failure of pancreatic exocrine function.

The nutritive status was calculated based on the measures of body mass, height, age, gender, and body mass index (BMI) according to Quetelet (mass (kg)/height (m)^2^) [38,39,40,41].

### 4.2. Determination of the E92K Genetic Variant

The determination of the E92K genetic variant is carried out by a multiplex system to analyze nine private Russian patients with variants in the *CFTR* gene, including the pathogenic variant E92K, using the allele-specific ligase reaction method (Figure 7) or by restriction analysis (Figure 8) using modified primers (F 5 ′ TCTGTTTTTCCCCTTTTGTCG; R 5 ′ ATTCTCATCTGCATTCCAAG) [42].

### 4.3. Methods for Evaluating the Functional Activity of the CFTR Channel

The method of determining the conductivity of sweat using the “Nanoduct” system (Wescor, South Logan, Utah, USA) was used as a sweat test. The assessment was carried out according to the recommendations of the European Consensus [43].

Patients with the genetic variant E92K, homozygotes F508del, and healthy took part in the study using the methods of ICM and intestinal organoids, all study participants were included in one of six groups. The quantitative composition and genotypes of the study participants are presented in Appendix A.

#### 4.3.1. Intestinal Current Measurements (ICM)

ICM method is based on determining the functional activity of ion channels, including chloride ones, and the rectal mucosa by the addition of stimulants (amiloride, forskolin/IBMX, genistein, carbachol, 4,4′-diisothiocyanatostilbene-2,2′-disulfonic acid (DIDS), and histamine). The study was carried out after signing the informed voluntary consent of the patient’s parents. The study with the ICM method was performed according to the European Standard Operating Procedures V2.7_26.10.11 (SOP) [44] and previously described in the authors’ article [18].

In the first stage, each of the four recirculation chambers was calibrated separately on the VCC MC 8B421 Physiological Instrument (San Diego, CA, USA). Physical factors were considered, such as the presence of air in the contact tips with agar and the resistance of the liquid, as well as environmental factors such as the absence of vibrations near the equipment, accidental contact with the electrodes, and the absence of extraneous working devices in the office. In the second stage, after the calibration of the device, the rectal biopsy material was placed in the chamber. Biopsy samples were collected using Olympus Disposable EndoTherapy EndoJaw Biopsy forceps (model #FB-23OU) according to the manufacturer’s instructions. The size of the biopsy sample was 3–5 mm. The biopsy material was placed in a special slider P2407B with a 1.2 mm diameter diaphragm, which was then inserted into the Ussing chambers. The chambers were filled with Meyler buffer solution. The buffer was prepared before the study and included the following: 105 mM NaCl, 4.7 mM KCl, 1.3 mM CaCl_2_•6H_2_O, 20.2 mM NaHCO_3_, 0.4 mM NaH_2_PO_4_•H_2_O, 0.3 mM Na_2_HPO_4_, 1.0 mM MgCl_2_•6H_2_O, 10 mM HEPES, and 10 mM D-glucose, as well as 0.01 mM indomethacin. The biopsies were heated to 37 °C using a circulation pump connected to a temperature-controlled water bath and continuously carbonated with 95% O_2_ and 5% CO_2_. The registration of the study began with the recording of the basal short-circuit current (preamiloride stage). At the third stage, stimulators (Sigma-Aldrich, Merck, Taufkirchen, Germany) were added in the following sequence: amiloride (100 mM), forskolin (10 mM)/IBMX (100 mM), genistein (100 mM), carbachol (100 mM), DIDS (100 mM), and histamine (100 mM). The study was completed after the basal short-circuit current was recorded. Stimulants were characterized as follows: amiloride inhibits ENaC (sodium channels), forskolin/IBMX (3-isobutyl-1-methylxanthine) activates cAMP-dependent chloride channels (CFTR), genistein activates the opening of the CFTR channel, carbachol initiates the opening of the Ca^2+^ (calcium channel), DIDS is an inhibitor of anionic transport through biological membranes, and histamine reactivates the Ca^2+^-dependent secretory pathway.

The control for the data obtained was the results of a multicenter study “Multicenter Intestinal Current Measurements in Rectal Biopsies from CF and Non-CF Subjects to Monitor CFTR Function” [27], as well as own comparison groups—13 healthy people without hereditary pathology and diseases of the pancreas and gastrointestinal tract (control group). For this method, patients were divided into groups according to their genotype: Group 1—E92K/E92K, Group 2—E92K/I class, Group 3—E92K/F508del, Group 4—E92K/IV-V classes, and Group 5—F508del/F508del (Appendix A). For the ICM method, each quantitative value represents the mean of three measurements (for each patient, the measurements were preformed from three biopsy samples).

#### 4.3.2. Human Intestinal Organoids Culture

Cultures of intestinal organoids were obtained in accordance with protocols developed in the Netherlands by a group of scientists led by J.M.Beekman [15,16,17,45].

Intestinal organoids were obtained by isolating crypts from rectal biopsies. For this purpose, rectal biopsies were washed with PBS (10 times by 10 mL) and then treated for 30 min with 10 mM EDTA in PBS. Isolated crypts were collected by centrifuging (130 g for 5 min at 4 °C) and then seeded in 50% “Matrigel” (Corning, Corning, NY, USA) into 24-well plates, as described in the work of Vonk A. et al. [45]. The organoids were passed ~ 1 time a week, updating the culture medium every 2–3 days [46]. Culture medium was prepared according to Vonk A. et al. [45].

#### 4.3.3. Forskolin-Induced Swelling (FIS) Assay

The forskolin-induced swelling assay was adapted from Dekkers et al. [16]. Notably, 24 h before FIS assay, the organoids were replated, where 7-day cultures of large budding organoids were mechanically dissociated into smaller parts by pipetting. Intestinal organoids were seeded in matrigel drops (4 μL drop with ~30 small organoids) into 96-well plates in 50 μL of culture medium with or without the addition of a 3.5 μM corrector VX-809 (Selleckchem, Houston, TX, USA). After 24 h, the organoids were incubated with 0.85 μM of Calcein green (Biotium, Fremont, CA, USA) for 20 min; immediately before visualization, they were stimulated only with forskolin (5 μM or 0.128 μM) (Sigma-Aldrich, St. Louis, MI, USA), or in combination with 3.5 μM of VX-770 (ivacaftor, Selleckchem, Houston, TX, USA). The organoids were analyzed using an Observer. D1 fluorescence microscope (Zeiss, Oberkochen, Germany) at 37 °C with 5% CO_2_ for 60 min, obtaining images every 20 min. The swelling of organoids caused by forskolin was determined quantitatively using the Sigma Plot 12.5 software (SYSTAT Software Inc., San Jose, CA, USA). The analysis of the swelling of organoids is expressed as the absolute area under the curve (AUC), calculated from the normalized increase in the surface area (baseline = 100%, t = 60 min). The quantitative assessment of the CFTR response to modulators in organoids was calculated as the difference between untreated organoids (Fsk) and treated organoids (VX-809 or VX-770 or VX-809+ VX-770). It was believed that there is an answer if the AUC was higher than 1000 after incubation of CFTR modulators, as described by Ramalho et al. [30]. Within each organoid experiment, every test condition was assessed in duplicate. Concurrently, each quantitative value of a test condition represents a mean of 4–6 replicates (from separate plate wells). For the method of intestinal organoids, patients were divided into groups according to their genotype (Appendix A).

### 4.4. RT-PCR Analysis

To assess the effect of E92K on pre-mRNA splicing, we analyzed mRNA obtained from the intestinal organoids culture of healthy control and E92K homo- and heterozygous E92K/F508del patients. Total RNA was isolated using ExtractRNA reagent (Eurogen, Moscow, Russia) and treated with DNAseI (Thermo Fisher, Carlsbad, CA, USA). cDNA was obtained using MMuLV H—reverse transcription system (DIALAT Ltd., Moscow, Russia). PCR was performed with the following primers: CFTR Ex2F (5′ CGCCTGGAATTGTCAGACATA 3′) and CFTR Ex6R (5′ CATCATTCTCCCTAGCCCAG 3′). The PCR products were analyzed by 2% agarose gel electrophoresis and Sanger sequencing.

### 4.5. NMD Inhibition

For the nonsense-mediated mRNA decay inhibition, the E92K/F508del patient’s intestinal organoids culture was incubated for 6 h with 300 μM cycloheximide. Then, mRNA was isolated, cDNA was obtained, and PCR was performed, as described above.

### 4.6. Targeted Next-Generation Sequencing of PCR-Product

For deep sequencing of RT-PCR products obtained from intestinal organoids of healthy control and E92K homozygous patients, we used next-generation sequencing. cDNA of heterozygous E92K healthy carrier was received from nasal epithelium cells obtained by brushing the nasal cavity with interdental brushes. After receiving cDNA, PCR was performed as described above. The obtained PCR fragments were taken for the library preparations using NEBNext^®^ Fast DNA Fragmentation & Library Prep Set for Ion Torrent (“New England Biolabs”, Ipswich, MA, USA) following manufacturer’s recommendations and sequenced on the Ion Torrent S5 (“Thermo Fisher Scientific”, Waltham, MA, USA) (with coverage > 200,000). The raw sequencing data were processed with a custom pipeline based on open-source bioinformatics tools HISAT2, Samtools, and StringTie. Splice junctions were visualized using Sashimi plot in IGV.

## 5. Conclusions

Russian and Turkish CF patients with the genetic variant E92K are characterized by an older age of diagnosis compared to homozygotes F508del and a high frequency of preservation of pancreatic function. However, the defeat of the bronchopulmonary system in this category of patients does not differ in severity from the group of homozygotes, according to F508del. The results of the sweat test and the ICM method showed partial preservation of the function of the CFTR channel. At the same time, the results of the FIS assay on cultures of intestinal organoids obtained from patients showed that the E92K variant causes serious violations of the function of the CFTR channel. Functional analysis of CFTR gene expression revealed a weak effect of the E92K variant on mRNA-CFTR splicing. In general, the study showed that the assignment of the E92K variant to a certain class of variants of the *CFTR* gene is difficult, establishing the need for an improved classification scheme for genetic variants. The results of the evaluation of the effect of CFTR modulators on intestinal organoids were positive—we showed that the response of organoids to the use of VX-809 and VX-809 + VX-770 is many times higher than the responses of F508del/F508del controls, which allows us to consider the E92K variant as a promising target for therapy with CFTR correctors.

## Figures and Tables

**Figure 1 ijms-24-06351-f001:**
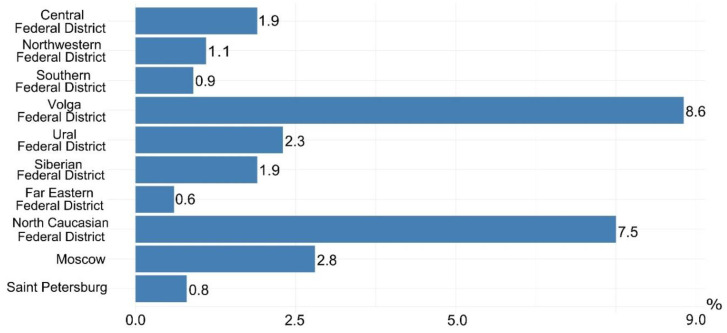
Allelic frequency of the E92K variant in the federal districts of the Russian Federation [5].

**Figure 2 ijms-24-06351-f002:**
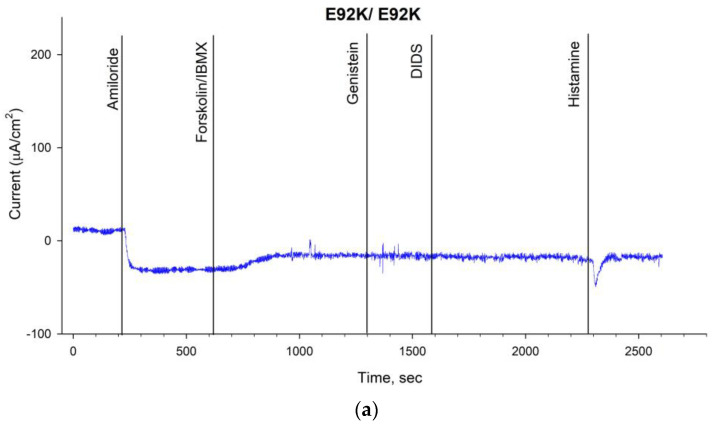
Intestinal current measurements (ICM) for patients carrying the genetic variant E92K; (**a**) E92K/E92K genotype (Group 1). With the introduction of amiloride, there was a decrease in the short−circuit current (ΔISC), in response to forskolin/IBMX, there is an increase in ΔISC, and a change in the short−circuit current in the negative direction was noted for the addition of histamine; (**b**) E92K/I class genotype (Group 2). With the introduction of amiloride, there was a decrease in the short−circuit current (ΔISC), in response to forskolin/IBMX, there is an increase in ΔISC, and a change in the short−circuit current in the negative direction was noted for the addition of histamine; (**c**) E92K/F508del genotype (Group 3). With the introduction of amiloride, there was a decrease in the short−circuit current (ΔISC), in response to forskolin/IBMX, there is an increase in ΔISC, and a change in the short−circuit current in the negative direction was noted for the addition of histamine; (**d**) E92K/IV−V classes genotype (Group 4). When amiloride is administered, the response is not fixed, in response to forskolin/IBMX, there is an increase in a decrease in the short−circuit current (ΔISC), and a change in the short−circuit current in the negative direction is noted for the addition of histamine; (**e**) F508del/F508del genotype (Group 5). When amiloride was administered, there was a decrease in the short−circuit current (ΔISC), but no changes were observed in response to the introduction of forskolin/IBMX, and a negative change in the short−circuit current was observed in response to the addition of histamine; (**f**) healthy individual (control). The addition of amiloride caused a decrease in ΔISC, there was a significant increase of ΔISC in response to forskolin/IBMX, while the addition of histamine led to a change of short−circuit current in the positive direction.

**Figure 3 ijms-24-06351-f003:**
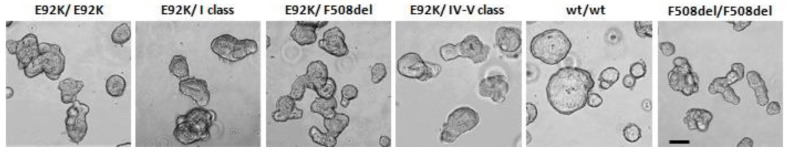
Morphological features of homo-/heterozygous organoids with the E92K-CFTR variant (groups 1–4) in comparison with F508del-homozygous and non-CF control (*wt*/*wt*) organoids. The scale bar is 200 microns.

**Figure 4 ijms-24-06351-f004:**
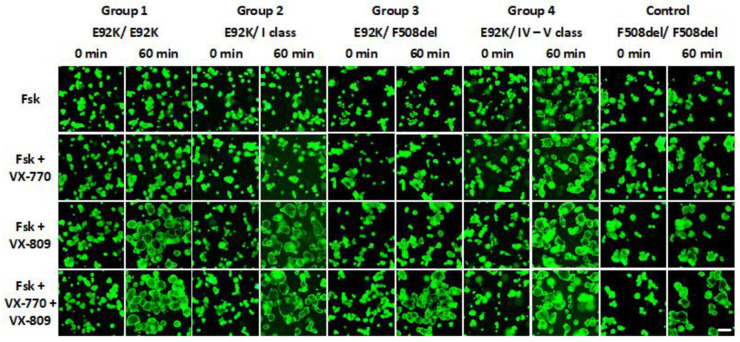
Investigation of the residual functional activity of the CFTR channel and the effect of the VX-770 potentiator and the VX-809 corrector on the rescue of function for homo-/heterozygotes according to the E92K-CFTR variant. Concentrations are 5 μM Fsk, 3.5 μM VX-770 and 3.5 VX-809. The scale bar is 200 microns.

**Figure 5 ijms-24-06351-f005:**
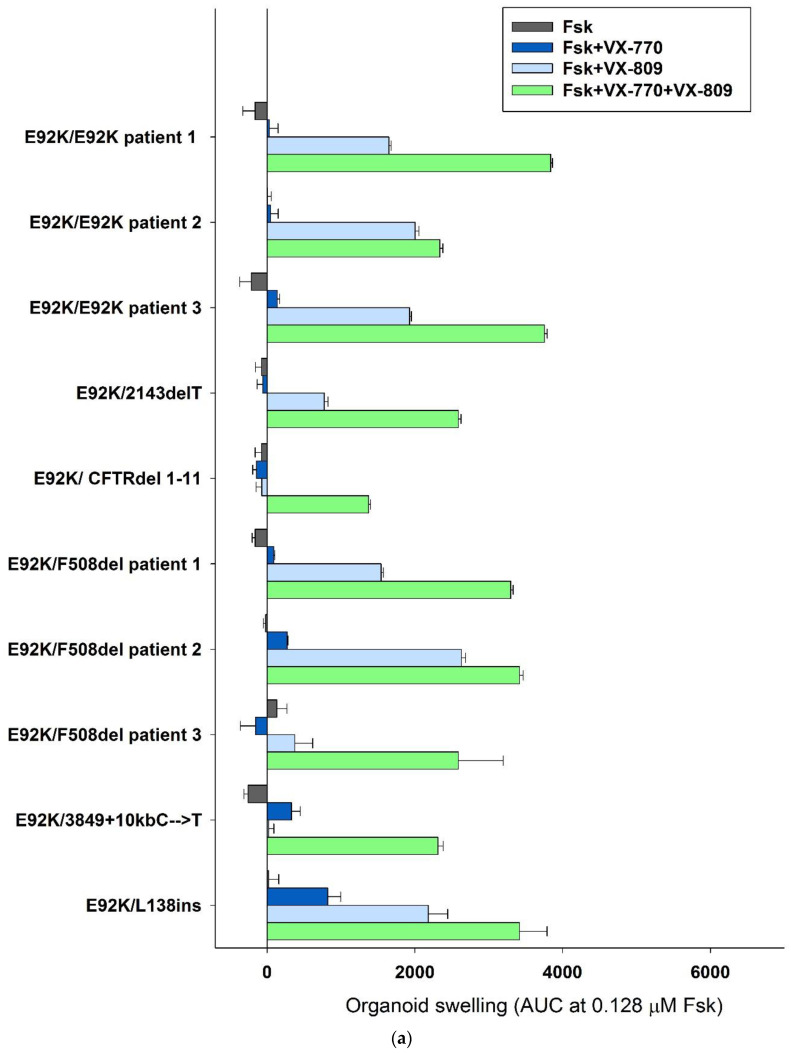
Results of quantitative assessment of organoid swelling under the action of forskolin and CFTR modulators for homo-/heterozygotes with the E92K-CFTR variant in comparison with F508del/F508del control. Concentrations—0.128 μM Fsk (**a**) and 5 μM Fsk (**b**); 3.5 μM VX-770 and 3.5 VX-809.

**Figure 6 ijms-24-06351-f006:**
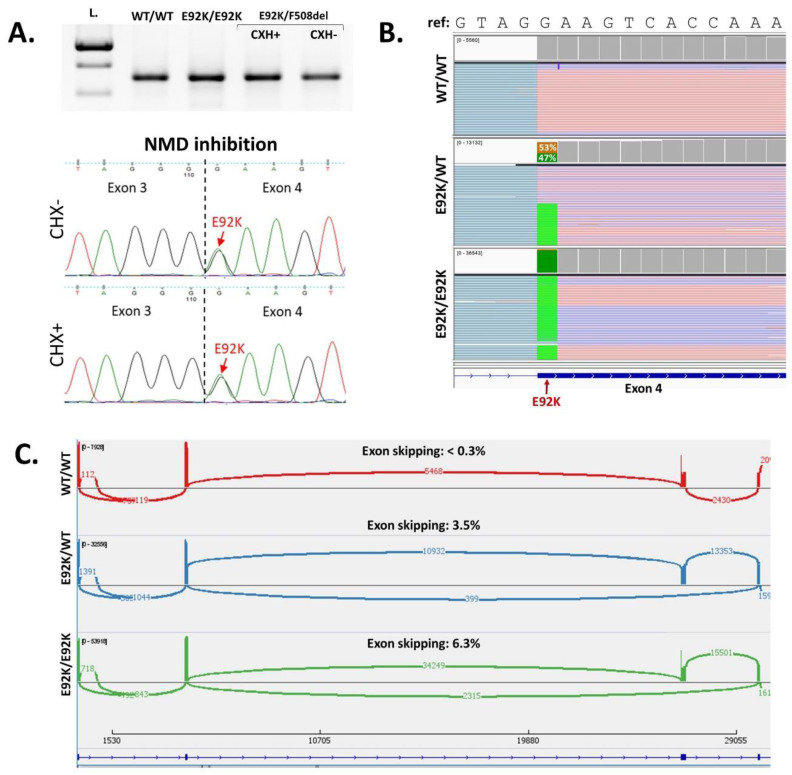
(**A**) Electrophoregram of RT−PCR products and sequencing of RT−PCR products obtained from intestinal organoids sample of heterozygous E92K/F508del patient before and after NMD inhibition; (**B**,**C**) targeted next−generation sequencing of PCR products covering *CFTR* exons 2–6 obtained from intestinal organoids samples of healthy control, E92K homozygous patient and healthy carrier of E92K variant; (**B**) analysis of nucleotide coverage around the E92K variant in the IGV browser. For each sample, the nucleotide coverage is presented above. Examples of the reads are depicted below; for the E92K/WT sample, the percentage of reads with each allele is shown; (**C**) analysis of splicing events using Sashimi plot in IGV browser.

**Figure 7 ijms-24-06351-f007:**
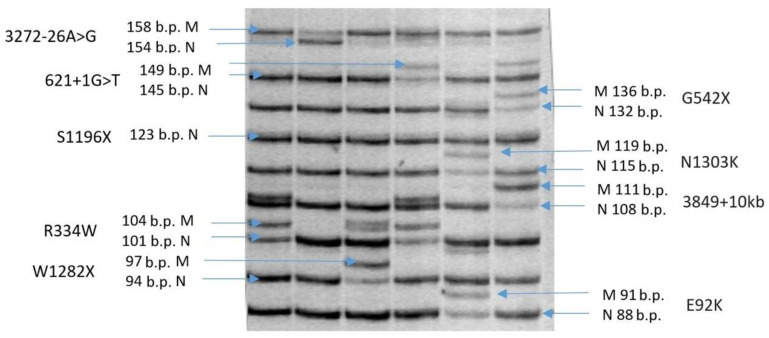
Fragment of an electrophoregram visualizing the detection system of the 9 most common pathogenic variants of the *CFTR* gene by allele-specific ligase reaction.

**Figure 8 ijms-24-06351-f008:**
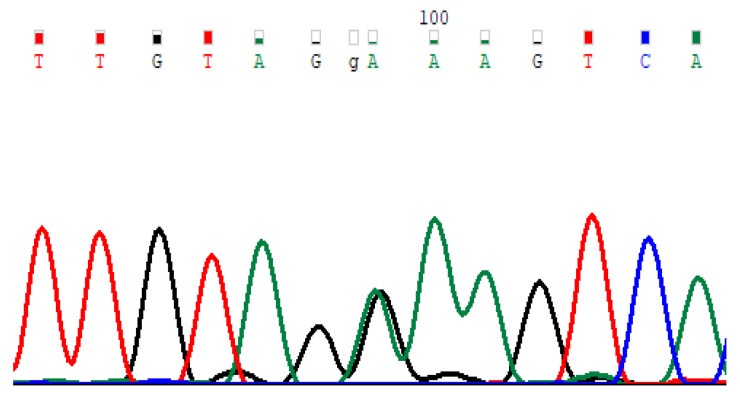
Chromatogram of 4 exon fragment sequences (c.274G > A).

**Table 1 ijms-24-06351-t001:** Clinical characteristics of cystic fibrosis patients with different genotypes.

			E92K/ E92K Group 1	E92K/ I Class Group 2	E92K/ F508del Group 3	E92K/IV-VI Classes Group 4	F508del/ F508del Group 5	*p*	*p*
Number of patients			33	36	82	8	881		
Age at the time of the study, years	Me (Q25;Q75)		12.1 (7.4; 22.5)	9.0 (4.7; 19.2)	13.6 (6.8; 22.7)	14.2 (11.1; 25.3)	9.7 (5.2; 15.6)	*p* = 0.004	P3–5 = 0.014
Age of diagnosis, years	Me (Q25;Q75)		0.8 (0.4; 11.4)	0.6 (0.2; 7.4)	2.7 (0.3; 9.5)	6.5 (4.2; 14.0)	0.3 (0.1; 1.3)	*p* < 0.001	P1–5 < 0.001 P2–5 = 0.028 P3–5 < 0.001 P4–5 = 0.001
Sweat test Conductivity, mmol/L			100.0 (90.5; 102.0)	105.8 (97.5; 114.5)	108.5 (93.0; 120.0)	92.0 (88.5; 111.0)	112.0 (101.0; 121.0)	*p* = 0.016	P1–5 = 0.019
Fecal elastase		≥200 ng/g	12 (75.0%)	12 (60.0%)	29 (76.3%)	4 (80.0%)	44 (9.8%)	*p* < 0.001	P1–5 < 0.001 P2–5 < 0.001 P3–5 < 0.001 P4–5 = 0.001
Diagnosis by newborn screening		Yes, positive,%	11 (34.4%)	11 (39.3%)	29 (36.3%)	1 (14.3%)	467 (54.7%)	*p* = 0.008	
		Yes, negative, %	0 (0.0%)	2 (7.1%)	2 (2.5%)	0 (0.0%)	15 (1.8%)		-
Microbiological examination		Chronic Ps. aeruginosa	9 (27.3%)	9 (25.7%)	30 (36.6%)	3 (37.5%)	295 (33.9%)	*p* = 0.738	-
		Intermittent Ps. aeruginosa	2 (6.1%)	7 (20.0%)	15 (18.5%)	1 (14.3%)	133 (15.7%)	*p* = 0.509	-
		Chronic Staph. aureus	20 (60.6%)	17 (48.6%)	41 (50.0%)	2 (25.0%)	526 (60.8%)	*p* = 0.053	-
		Chronic Burkholderia cepacia complex	2 (6.1%)	2 (5.7%)	9 (11.0%)	1 (12.5%)	59 (6.8%)	*p* = 0.502	-
		H. influenzae	1 (3.1%)	1 (2.9%)	1 (1.30%)	0 (0.0%)	52 (6.2%)	*p* = 0.405	-
		Achromobacter	1 (3.1%)	2 (5.6%)	4 (4.9%)	0 (0.0%)	48 (5.6%)	*p* > 0.99	-
		MRSA	3 (9.4%)	0 (0.0%)	6 (7.3%)	2 (25.0%)	30 (3.5%)	*p* = 0.010	P4–5 = 0.008
Respiratory function		FEV1, %	80.3 ± 31.3	73.6 ± 33.7	71.2 ± 30.0	72.8 ± 21.6	79.3 ± 24.9	*p* = 0.271	
		FVC, %	83.8 ± 30.1	79.3 ± 28.4	77.7 ± 24.2	87.0 ± 8.3	86.8 ± 20.7	*p* = 0.153	
BMI, kg/m^2^ Me (Q25;Q75))			16.0 (14.4; 19.5)	17.0 (15.1; 18.8)	17.2 (14.9; 19.8)	18.0 (14.5; 19.3)	15.8 (14.6; 17.8)	*p* = 0.004	P3–5 = 0.011
Complications		ABPA	0 (0.0%)	2 (5.7%)	3 (3.7%)	0 (0.0%)	14 (1.6%)	*p* = 0.176	
		Diabetes (insulin treatment)	0 (0.0%)	0 (0.0%)	0 (0.0%)	0 (0.0%)	30 (3.4%)	*p* = 0.363	
		Hemoptysis	1 (3.0%)	0 (0.0%)	4 (4.9%)	0 (0.0%)	8 (0.9%)	*p* = 0.046	P3–5 = 0.006
		Occur malignancy	0 (0.0%)	0 (0.0%)	0 (0.0%)	0 (0.0%)	1 (0.1%)	*p* > 0.99	
		Osteoporosis	1 (5.9%)	1 (3.6%)	4 (6.8%)	0 (0.0%)	46 (7.7%)	*p* = 0.976	
		Polyposis of the upper respiratory tract	4 (12.1%)	7 (20.0%)	25 (30.5%)	1 (12.5%)	253 (30.7%)	*p* = 0.090	
Liver damage		Cirrhosis of the liver with hypertension	0 (0.0%)	1 (100%)	2 (33.3%)	0 (0.0%)	44 (21.4%)	*p* = 0.566	
		Cirrhosis of the liver without hypertension	0 (0.0%)	0 (0.0%)	1 (16.7%)	0 (0.0%)	24 (11.7%)		
		Liver damage without cirrhosis	1 (100%)	0 (0.0%)	3 (50.0%)	0 (0.0%)	138 (67.0%)		
Therapy		NaCl	15 (45.5%)	18 (51.4%)	53 (64.6%)	3 (37.5%)	627 (72.0%)	*p* < 0.001	P1–5 = 0.010
		Antibiotic	7 (21.2%)	12 (33.3%)	37 (45.1%)	4 (50.0%)	417 (47.7%)	*p* = 0.021	P1–5 = 0.028
		Antibiotic Intravenous	8 (25.0%)	12 (33.3%)	27 (32.9%)	2 (25.0%)	342 (39.3%)	*p* = 0.321	
		Antibiotic tablets	11 (33.3%)	17 (47.2%)	45 (54.9%)	5 (62.5%)	540 (61.8%)	*p* = 0.006	P1–5 = 0.010
		Bronchodilator	9 (27.3%)	16 (44.4%)	39 (47.6%)	5 (62.5%)	444 (50.8%)	*p* = 0.087	
		Oxygen	0 (0.0%)	3 (8.3%)	6 (7.3%)	0 (0.0%)	28 (3.2%)	*p* = 0.112	
		rhDNase	31 (93.9%)	35 (97.2%)	81 (98.8%)	8 (100%)	852 (97.1%)	*p* = 0.529	
		Azithromycin	4 (12.1%)	6 (16.7%)	14 (17.1%)	3 (37.5%)	287 (33.2%)	*p* = 0.001	P3–5 = 0.028
		Ursodeoxycholic acid	24 (75%)	30 (83.3%)	66 (80.5%)	5 (62.5%)	820 (93.5%)	*p* < 0.001	P1–5 = 0.001 P3–5 < 0.001 P4–5 = 0.005
		Pancreatic enzymes	24 (75%)	24 (66.7%)	63 (76.8%)	5 (62.5%)	869 (99.2%)	*p* < 0.001	P1–5 < 0.001 P2–5 < 0.001 P3–5 < 0.001 P4–5 < 0.001
		PPI	2 (6.1%)	4 (11.8%)	17 (20.7%)	0 (0.0%)	175 (20.1%)	*p* = 0.118	
		Vitamin	21 (65.6%)	34 (94.4%)	61 (76.3%)	7 (87.5%)	827 (95.1%)	*p* < 0.001	P1–2 = 0.026 P1–5 < 0.001 P3–5 < 0.001
		CF physiotherapy	23 (71.9%)	23 (63.9%)	60 (74.1%)	7 (87.5%)	741 (85.7%)	*p* < 0.001	P2–5 = 0.004

**Table 2 ijms-24-06351-t002:** Indicators of short-circuit current density (ΔISC) in response to the introduction of stimulants in patients with the genetic variant of E92K in the genotype.

	**E92K/E92K (Group 1)**					
patient	ΔISC, µA/cm^2^	amiloride	forskolin/IBMX	genistein	DIDS	histamine
1	M ± m CF	−23.5	8.75 ± 3.89	1.5 ± 0.71	1.5 ± 0.71	18.25 ± 5.3
2	M ± m CF	−0.5	7.83 ± 2.13	0.5	0.5	12.5 ± 3.37
3	M ± m CF	−4.67 ± 3.01	5.33 ± 1.02	0.5	0.5	6.33 ± 0.82
M ± m total Me		−7.81 ± 3.83 2.5	7.13 ± 1.05 6.5	0.75 ± 0.2 0.5	0.75 ± 0.2 0.5	11.63 ± 2.26 10.75
	**E92K/CFTRdele1-11 и E92K/2143delT (Group 2)**					
	ΔISC, µA/cm^2^	amiloride	forskolin/IBMX	genistein	DIDS	histamine
1	M ± m CF	−5.5 ± 1.87	6.67 ± 2.68	1	0.5	53.83 ± 10.51
2	M ± m CF	−6.33	4.67 ± 1.02	1.17 ± 0.2	1.17 ± 0.2	13.17 ± 6.42
M ± m total Me		−5.92 ± 1.01 −5.75	5.67 ± 1.25 5.25	1.08 ± 0.09 1	0.83 ± 0.18 0.75	33.5 ± 11.11 30.75
	**E92K/F508del (Group 3)**					
	ΔISC, µA/cm^2^	amiloride	forskolin/IBMX	genistein	DIDS	histamine
1	M ± m CF	−8 ± 0.71	17.75 ± 3.89	0.75 ± 0.35	1	7.25 ± 1.77
2	M ± m CF	−4.5 ± 2.12	17.75 ± 3.18	1.75 ± 0.35	1.75 ± 0.35	2.25 ± 0.35
3	M ± m CF	−38.83 ± 4.32	17.17 ± 1.43	1	1	17.33 ± 2.68
M ± m total		−20.21 ± 7.29	17.5 ± 0.97	1.14 ± 0.19	1.21 ± 0.16	10.14 ± 3.02
	**E92K/L138ins и E92K/3849 + 10kbC > T (Group 4)**					
	ΔISC, µA/cm^2^	amiloride	forskolin/IBMX	genistein	DIDS	histamine
1	M ± m CF	−5 ± 0.61	14.5 ± 1.87	2.5	2.5	37.33 ± 3.56
2	M ± m CF	−4.67 ± 0.74	5.83 ± 1.63	2.5	2.5	14.83 ± 2.35
M ± m total		−4.83 ± 0.39	10.17 ± 2.34	2.5	2.5	26.08 ± 5.77
M ± m F508del/F508del		−18.39 ± 5.62	3.06 ± 0.89	1.83 ± 0.35	1.83 ± 0.35	21.5 ± 5.46
M ± m PI-CF *		−23.67 ± 4.36	2.97 ± 0.61	1.4 ± 0.25	1.67 ± 0.28	19.07 ± 3.69
Healthy M ± m		−8.98 ± 2.23	25.78 ± 3.37	2 ± 0.2	1.8 ± 0.18	101.68 ± 9.78
Reference value according to European SOP *		−8.5 ± 10.7	19.5 ± 13.4	-	-	32.4 ± 19.7

Note: ΔISC—short-circuit current, PI-CF—cystic fibrosis with pancreatic insufficiency, *—data from a multicenter study of patients with nucleotide sequences of the *CFTR* gene.

**Table 3 ijms-24-06351-t003:** Frequency of genotypes containing the E92K variant.

Genotypes (Legacy Name)	Genotypes (c. DNA Name)	Genotypes (Protein Name)		Gender		Total	Russian	Turkish
				Male	Female			
E92K/E92K (Group 1)	(c.[274G > A];[274G > A]	p.[Glu92Lys];[Glu92Lys]	N	14	19	33	25	8
			%	42.4	57.6	100.0		
E92K/I class (Group 2)	c.[274G > A];[I class]	p.[Glu92Lys];[I class]	N	21	15	36	33	3
			%	58.3	41.7	100.0		
E92K/F508del (Group 3)	c.[274G > A];[1521_1523delCTT]	p.[Glu92Lys];[Phe508del]	N	37	45	82	81	1
			%	45.1	54.9	100		
E92K/IV-VI classes (Group 4)	c.[274G > A];[IV-V classes]	p.[Glu92Lys];[IV-V classes]	N	3	5	8	6	2
			%	37.5	62.5	100.0		
F508del/F508del (Group 5, control)	c.[1521_1523delCTT];[1521_1523delCTT]	p.[Phe508del];[Phe508del]	N	442	439	881	787	94
			%	50.2	49.8	100.0		

## Data Availability

The datasets used and/or analyzed during the current study are available from the corresponding author upon reasonable request.

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
