# Peer review of "Clinical and Functional Characteristics of the E92K CFTR Gene Variant in the Russian and Turkish Population of People with Cystic Fibrosis"

_ijms, 2023, doi:10.3390/ijms24076351_

Round 1

Author Response

Response to Reviewer 1 Comments

Major corrections

  1.  The small number of patient samples, except wildtype, for the ISC and organoid measurements are a serious shortcoming. Furthermore, it is not clear how many times these experiments (ISC and FIS organoids) were performed from patient samples (n=3 or more?).

Response 1: FIS method based on intestinal organoids provides personalized approach for studying the residual CFTR channel function, which implies that even results obtained from a single patient are significant. E.g. Beekman JM et al. (doi: 10.1126/scitranslmed.aad8278; doi: 10.1183/13993003.02426-2019) provided results of organoid swelling for many rare CFTR variants obtained from single patients.

Within each organoid experiment, every FIS assay condition was assessed in duplicate. Concurrently, each quantitative value of a single FIS assay represents mean of 4-6 replicates (from separate plate wells). For ICM method each quantitative value represents mean of three measurements (for each patients the measurements were preformed from three biopsy samples).

The corresponding changes were included in the revised manuscript

  1. Missing CFTR inhibitor control for ISC measurements (Figure 2). Authors used DIDS but this is not specific for CFTR. Should use CFTRinh-172 to calculate percentage contribution of CFTR compared to total chloride channels in samples.

Response 2: ICM studies were performed according to standard european operational protocols V2.7_26.10.11 approved by ECFS Diagnostic Network Working Group in 2011. CFTRinh-172 is not included in the standard algorithm.

  1. For organoid swelling experiments, missing wildtype as an important control (Figures 4 & 5) to compare to samples from CF patients. Wildtype is shown in figure 3, the authors need to perform experiments with wildtype included (Figures 4 & 5)

Response 3: Wt/wt organoids are not employed in FIS assay as controls, because they already possess a wide lumen due to a functional CFTR channel and thus show insignificant swelling upon forskolin treatment in essence. Therefore, organoid lines obtained from CF patients are used as controls  in FIS assay. In this case internal lumen is reduced due to impaired CFTR-channel function, whil upon CFTR modulator treatment this function is restored and forskolin stimulation causes organoid swelling. Thus, Vonk AM et al. (Vonk AM, van Mourik P, Ramalho AS, Silva IAL, Statia M, Kruisselbrink E, Suen SWF, Dekkers JF, Vleggaar FP, Houwen RHJ, Mullenders J, Boj SF, Vries R, Amaral MD, de Boeck K, van der Ent CK, Beekman JM. Protocol for Application, Standardization and Validation of the Forskolin-Induced Swelling Assay in Cystic Fibrosis Human Colon Organoids. STAR Protoc. 2020 Jun 3;1(1):100019. doi: 10.1016/j.xpro.2020.100019. PMID: 33111074; PMCID: PMC7580120, p. 5) used multiple CF organoid lines as controls including F508del/F508del, that we accordingly used as cotrol in our study.

  1. Authors organoid data contradict the clinical data, and the authors provide several explanations in their discussion for possible sources of the contradiction. One source suggested is alteration of gene expression and protein synthesis during the two-week cultivation. It would be beneficial and not difficult for the authors to measure changes in CFTR gene expression and other associated genes (e.g., ENAC or other Chloride channels) to test this hypothesis.

Response 4: We have obtained a result that requires further research and may be repeated with other genetic variants. Estimation of expression changes of CFTR and other associated genes was not aimed in our study. We plan to continue the research of E92K variant and perform the corresponding analyses in our future studies and articles. We thank the dear reviewer for this point!

Minor corrections

  1. Please provide literature or database reference for the following statement “so the frequency of the variant 2183AA→G in the population of Armenia is 70 10.0%, and the variant R1162X in Slovenia is 5.0%.“ found on line 70 in the introduction.

Response 1: Corrected

  1. On line 64 & 79, “can be attributed to frequent “ could be shortened to “are frequent in” for better grammar.

Response 2: Corrected. Thank you!

  1. In table 1 & 3 and text please define or write out terms Me and M. abbreviations could be confusing for readers

Response 3: Corrected. Thank you!

  1. Adjust first column width in table 1 to not cutoff words onto second line for easier readability.

Response 4: Changed

  1. Missing hyphen “group 3 E92K/F508del” line 176

Response 5: Corrected

  1. Figure 5 patient misspelled twice as “pacient” third and fourteenth from bottom of graph

Response 6: Corrected, thank you!

  1. In figure 6B, information is not clear in the screenshot. The y-axis of plots are not labeled. It is unclear what information the reader should obtain from this part of figure. Add more information in legend and reformat 6B for clarity and labeling

Response 7: The missing information was added in the revised figure

  1. Please provide AUC data for FIS measurements in supplemental data

Response 8: The missing data was added.

  1. Please provide more details on “NGS libraries were prepared and sequenced on the 567 Ion Torrent S5 (with coverage > 200’000).” How the libraries were prepared and sequenced

Response 9: The necessary information was included (The obtained PCR fragments were taken for the library preparations using NEBNext® Fast DNA Fragmentation & Library Prep Set for Ion Torrent (“New England Biolabs”, USA) following manufacturer's recommendations and sequenced on the Ion Torrent S5 (“Thermo Fisher Scientific”, USA) (with coverage > 200’000).

Response 9: Added.

  1. Reference 8, please provide year of registry report

Response 10: Added

  1. Please provide English translation for reference 10

Response 11: The missing text was included

  1. Reference 36, changing formatting to match other references (it is italics)

Response 12: Corrected. Thank you!.

  1. Please double check link to CFTR database in references, it did not work properly

Response 13: Corrected. Thank you!.

  1. Please provide more detail information for methods 4.3.1, 4.3.2, 4.3.3 so other researchers could reproduce experiments

Response 14: Added.

Reviewer 2 Report

Table 1 is difficult to read, consider to move to a supplementary table.

Consider to shorten the introduction and discussion section

Author Response

Response to Reviewer 2 Comments

  • Table 1 is difficult to read, consider to move to a supplementary table

We have significantly reduced the table, it is one of the main results of our study and provides important information for physicians caring for patients with CF

  • Consider to shorten the introduction and discussion section

We have tried to reflect the description of a rare genetic variant of the CFTR gene in the introduction and discussion in order to expand the knowledge of specialists. In addition, other reviewers recommended expanding it and we edited it minimally.

Reviewer 3 Report

Thank you for your interesting submission and work.

This is a study  evaluating dysfunction of the CFTR channel in patients bearing the E92K variant in their genotype. The study also presents interesting clinical data on the phenotype presented by these patients. In my opinion, this paper can be improved with some minor revisions.

I hope my comments may help to improve the paper

Minor issues:

Key words:  it would be useful to include the words “CF Registry” and “genotype-phenotype correlation”

Intro: in my opinion, the description of the frequency of the E92K variant (results 2.1) and the genotype-phenotype correlation (results 2.2) should be present among the aims of the study

Results:

Table 1: To an easier reading and a better understanding of the numerous data presented, I would suggest reducing the information in Table to the essentials and presenting the complete data as supplementary material.

1. please, add the number of patients in the five groups studied

2. please, present ages and sweat test values with median value, range and IQR25;IQR75; eliminating the means, especially when the SDs are greater than the mean value

3. please, present only pancreatic insufficiency data (fecal elastase < 200)

4. please, check the data of  numbers and percentages of meconium ileus line: do they refer to the absence of meconium ileus?

5. please, submit only positive and negative newborn screening data *

And. please, in table and in text (lin 461), correct to FEV1 % of predicted, FVC % of predicted

6. microbiological examination/complications/therapies: insert in the table only the data that present a significant difference

Lin 165: I would change the term "soft" to the term "mild" in all the text

Table 2: I suggest moving it to the supplementary material

Material and Methods:

Lin.466 and following: The Authors explain how to evaluate the nutritional status in CF patients, children and adults (as BMI centile). However, in Table 1 only BMI (kg/m2) is reported and no difference by ages. It could be better to modify the Table, reporting BMI centiles (instead of BMI) for pediatric population  and adding the number and  % of malnourished subjects in each group.

* Why there is a so great difference in % of screening positive subjects among the five groups? Does it depend on different birth-regions? Please, clarify

This may be an explanation for the different age at diagnosis among groups. Furthermore, lung damage in subjects with E93K could be consequent to a more delayed diagnosis than in the group of F508del homozygotes, in which more than half of the subjects received an early diagnosis due to a positive newborn screening.

Please, the Authors could comment on these aspects in the discussion section.

Author Response

Response to Reviewer 3 Comments

Thank you for your careful reading of the manuscript!

  1. Key words:it would be useful to include the words “CF Registry” and “genotype-phenotype correlation”

Response 1: Added

  1. Intro: in my opinion, the description of the frequency of the E92K variant (results 2.1) and the genotype-phenotype correlation (results 2.2) should be present among the aims of the study

Response 2: The purpose of the study was added: “The aim of this study is to investigate dysfunction of the CFTR channel in patients with CF from the Russian Federation and Turkey, who have a common for these countries pathogenic variant E92K in the genotype, a frequency of this variant and the genotype-phenotype correlation»

  1. Table 1: To an easier reading and a better understanding of the numerous data presented, I would suggest reducing the information in Table to the essentials and presenting the complete data as supplementary material (The table was reduced, the remaining indicators comply with the requirements of the European Register for assessing the health status of patients with CF).
  2. please, add the number of patients in the five groups studied (Added)
  3. please, present ages and sweat test values with median value, range and IQR25;IQR75; eliminating the means, especially when the SDs are greater than the mean value (removed extra lines)
  4. please, present only pancreatic insufficiency data (fecal elastase < 200) (removed extra lines)
  5. please, check the data of  numbers and percentages of meconium ileus line: do they refer to the absence of meconium ileus? (removed, thanks)
  6. please, submit only positive and negative newborn screening data (deleted data with undefined cases)*

* Why there is a so great difference in % of screening positive subjects among the five groups? Does it depend on different birth-regions? Please, clarify. This may be an explanation for the different age at diagnosis among groups. Furthermore, lung damage in subjects with E93K could be consequent to a more delayed diagnosis than in the group of F508del homozygotes, in which more than half of the subjects received an early diagnosis due to a positive newborn screening. Please, the Authors could comment on these aspects in the discussion section.

The largest number of patients who were screened belongs to the group of F508del homozygotes, due to the fact that they are younger compared to groups I-IV, and screening in the Russian Federation began in 2007. Additions were made to the discussion section.

And. please, in table and in text (lin 461), correct to FEV1 % of predicted, FVC % of predicted (corrected, thanks)

  1. microbiological examination/complications/therapies: insert in the table only the data that present a significant difference (partially reduced)

  1. Lin 165: I would change the term "soft" to the term "mild" in all the text

Response 4: Replaced throughout the text

  1. Table 2: I suggest moving it to the supplementary material

Response 5: We have moved Table 2 it to the supplementary material, (Table S1)

  1. Material and Methods: Lin.466 and following: The Authors explain how to evaluate the nutritional status in CF patients, children and adults (as BMI centile). However, in Table 1 only BMI (kg/m2) is reported and no difference by ages. It could be better to modify the Table, reporting BMI centiles (instead of BMI) for pediatric population and adding the number and  % of malnourished subjects in each group.

Response 6: In order not to increase the volume of the article, we removed the information on the BMI assessment by percentiles from the methods and Table 1 shows the absolute values.

Round 2

Reviewer 1 Report

The authors revisions are acceptable and my concerns/edits have been addressed.